# Effect of Shoulder Fillet Radius on Welds in Bobbin Tool Friction Stir Welding of A1050

Huilin Miao [1], Takuya Miura [1,*], Wei Jiang [2], Masato Okada [2] and Masaaki Otsu [2]

1   Joining and Welding Research Institute, Osaka University, Osaka 567-0047, Japan
2   Department of Mechanical Engineering, University of Fukui, Fukui 910-8507, Japan
*   Correspondence: miura@jwri.osaka-u.ac.jp

**Abstract:** In this study, five bobbin tools with different shoulder fillet radii were employed for the bobbin tool friction stir welding (BT-FSW) of A1050-O sheets to systematically evaluate the effects of shoulder fillet radius on the welding defect formation, flash formation, weld thickness, grain size of the stir zone, and tensile properties. The quality classifications of the joints' appearance were summarized as process windows, and the appropriate welding condition range for each shoulder fillet radius was clarified. It was observed that an increase in the shoulder fillet radius decreased the welding defects and flash formation; however, it increased the minimum thickness of the weld except when the shoulder fillet radius was 0.5 mm. The grain size of the stir zone increased with increasing shoulder fillet radius from 0.5 mm to 6 mm. The ultimate tensile strength (UTS) of the stir zone decreased with increasing shoulder fillet radius from 0.5 mm to 1 mm, increased from 1 mm to 3 mm, and remained constant from 3 mm to 6 mm. The results indicate that a shoulder fillet radius larger than 3 mm is effective in decreasing flash formation and maintaining a constant weld thickness.

**Keywords:** bobbin tool friction stir welding; tool geometry; welding defect; flash formation; aluminum; tensile property



## 1. Introduction

Friction stir welding (FSW) is a solid-state welding technique that has many advantages over conventional fusion welding. According to the review by Mishra and Ma [1], FSW is an energy-efficient, environmentally friendly technology that is considered to be the most significant development in welding in recent years. Therefore, it has gained wide interest from both industry and academia. FSW has been considered for application to various materials (Al, Mg, Cu, Fe, Ti, and its alloys, etc.). In addition, varioues aspects of process optimization are also being studied. Liu et al. reported that the softening and tensile properties of A1050-H24 aluminum sheets are significantly affected by FSW parameters, such as the welding speed and tool rotation speed [2]. Fujii et al. studied the effects of carbon content and phase transformation on the mechanical properties and microstructures of carbon steel welded using FSW, which was the first example of welding general steels without phase transformation [3]. Wang et al. developed a novel friction stir spot welding technique using double-sided tools with adjustable probes that can obtain a flat joint without a keyhole in the welding of magnesium alloys and low-carbon steel [4,5]. Various novel metal-forming and welding methods have been developed by researchers based on the FSW concept. For example, Otsu et al. developed a friction stir incremental forming method by combining FSW and incremental sheet metal forming to form aluminum alloy sheets without heating from an external heat source [6].

In conventional FSW, since the stirring at the bottom of the joint along the plate thickness direction is relatively weak, kissing bonds or root defects may occur due to insufficient stirring and heat input at the bottom [7,8]. It is difficult to install a back support, making it difficult to apply conventional FSW to large hollow structural parts. In order

to solve these problems, bobbin tool friction stir welding (BT-FSW) was developed as a derivative technology of FSW, in which the material is sandwiched with shoulders from both sides. In some literature, it is also called self-support FSW or self-reacting FSW [9–22]. A schematic illustration of BT-FSW, which is an FSW variant, is shown in Figure 1. The bobbin tool consists of a probe, top shoulder, and bottom shoulder. BT-FSW has various advantages, such as the omission of the backing bar and an absence of root defect formation. Furthermore, in BT-FSW, since the tool load can be canceled by the internal stress of the tool, the rigidity required for the device and fixture is low, and the heat input is relatively high, so the material tends to soften, so the welding speed can be increased. It is possible to improve the efficiency compared to conventional FSW [9,10]. Zhou et al. explained that flash was formed due to heat accumulation at the retreating side in magnesium alloy FSW using a bobbin tool with upper and lower shoulders of different diameters [11].

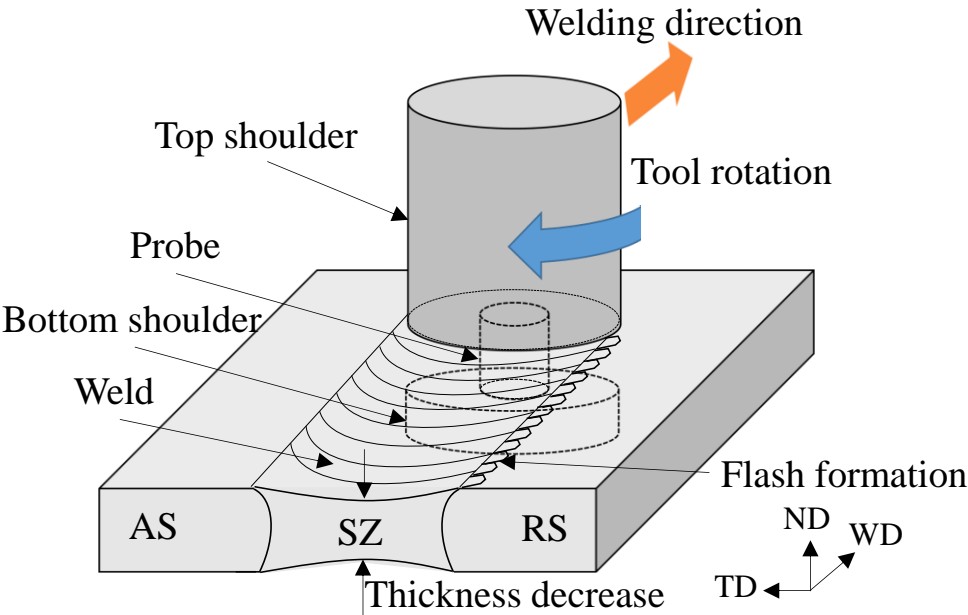

**Figure 1.** Schematic illustration of BT-FSW.

Because the shoulder and probe have a significant influence on the heat generation and material flow during both the FSW and BT-FSW processes, tool geometry is considered an influential aspect of the process development. Regarding the probe shape design, Reza-E-Rabby et al. investigated the effect of probe thread shapes on the weldability of different aluminum alloys (AA 7050 and AA 6061) with a cylindrical probe tool with four thread pitches, including an unthreaded probe [22]. It was observed that thread shapes were beneficial for improving the tool performance by inhibiting wormhole defects through effective material transportation. In contrast, Thomas et al. reported that a probe with a non-circular cross-section is effective for joining thick plates, and a probe with a spiral grooved shape is suitable for high-speed joining [23].

Regarding shoulder geometry, although a simple shoulder shape, such as a flat shoulder, can be manufactured easily, a flat surface shoulder often leads to excessive flash formation, as Unfried-Silgado et al. reported on AA1100 aluminum [24]. Casalino et al. also investigated the effects of shoulder geometry on weldability [25] and reported that a flat shoulder is sensitive to the process parameters and results in flash and welding defect formation. In contrast, a convex shoulder design can increase the contact area with the workpiece and facilitate the joining of workpieces of different thicknesses, as Nishihara and Nagasaka reported [26]. A concave shoulder design is commonly used to prevent material spilling and compress material around the probe during FSW, as demonstrated by Scialpi et al. [27] and reviewed by El-Moayed et al. [28].

Because the BT-FSW tool has two shoulders rather than one, the role of the shoulder geometry in BT-FSW is more significant than that in FSW. However, there are few studies on the BT-FSW shoulder shape design [10]. Therefore, the relationship between shoulder characteristics and welding defects needs further investigation. Moreover, flash is always formed as the weld thickness decreases, which results in welding defects and a decrease in strength, as reported by Li et al. [8]. Especially in BT-FSW, the two shoulders are plunged from both sides of the sheet at the same time, and the problems of flash formation and sheet thickness reduction are much more obvious than in FSW. As Galvão et al. indicated, most cylindrical tool shoulders require a 1–3° tool tilting angle to maintain the material reservoir and produce a compressive forging force on the weld [29]. However, it is difficult to utilize the tilt angle in BT-FSW because the bottom shoulder strikes the sheet when the bobbin tool is tilted.

De Giorgi et al. reported that a flat shoulder with a fillet radius of 1 mm results in negligible flash formation [30]. Recently, Jiang et al. conducted a BT-FSW experiment using a bobbin tool with a relatively large fillet radius of 6 mm at the edge of the shoulder with a total plunging depth of 0.2 mm and reported that a sound joint was obtained, and almost no flash formation was observed [31]. Other studies also used welding tools with relatively large radius fillets on the shoulder edges [12–16,27]. However, the effects of changes in shoulder fillet radius on flash formation and welding defects have not yet been systematically evaluated. Therefore, it is difficult to obtain a practical process window for the BT-FSW that can be used as a guideline for selecting welding parameters and a designing tool.

Considering the above data, in this study, the effect of the shoulder fillet radius ($R_{sf}$) on welding defects, flash formation, and weld thickness reduction in BT-FSW was analyzed. The BT-FSW was performed using tools with five different shoulder fillet radii. Flash formation, welding defects, the minimum thickness of the weld, the grain size of the stir zone (SZ), and tensile properties were systematically evaluated to elucidate the effects of the shoulder fillet radius on welds.

## 2. Materials and Methods

Pure aluminum sheets of JIS A1050-O with dimensions of 200 mm × 200 mm × 2.0 mm were used in this study. Stir-in-plate welding was performed using a three-axis vertical machining center (MILLAC 44V II, OKUMA Corp., Aichi, Japan). The apparatus is shown in Figure 2. The aluminum sheet was clamped to the table with a blank holder and screw bolts. A bobbin tool was introduced from a hole in the table into the sheet and removed from the hole after the FSW. A schematic of the bobbin tool used in the experiments is shown in Figure 3. The tool sizes used are listed in Table 1. Five bobbin tools with shoulder fillet radii ($R_{sf}$) of 0.5, 1, 2, 3, and 6 mm were prepared. A tool with a shoulder fillet radius of 1 mm is often used as a filleted tool [30]. The shoulders were simply flat to clarify the influence of the shoulder fillet radius. The diameter of the flat plane of the shoulder ($F_s$) was kept constant at Φ8 mm for all tools, which means that the total diameter of the shoulders varied between Φ9 mm, Φ10 mm, Φ12 mm, Φ14 mm, and Φ20 mm with changes in $R_{sf}$. The tools were made of stainless steel (JIS SUS304), which is easily available and has a good balance of high-temperature strength and workability. M6 right-hand screw bolts were used to connect the top and bottom shoulders, and the gaps between the shoulders were adjusted to a constant value of 1.8 mm, which is 0.2 mm smaller than the sheet thickness, using a feeler gauge. The welding speed and tool rotation speed were varied from 1000 to 5000 mm/min and 1000 to 7000 rpm, respectively, to evaluate the effect of the welding parameters on the weld morphology.

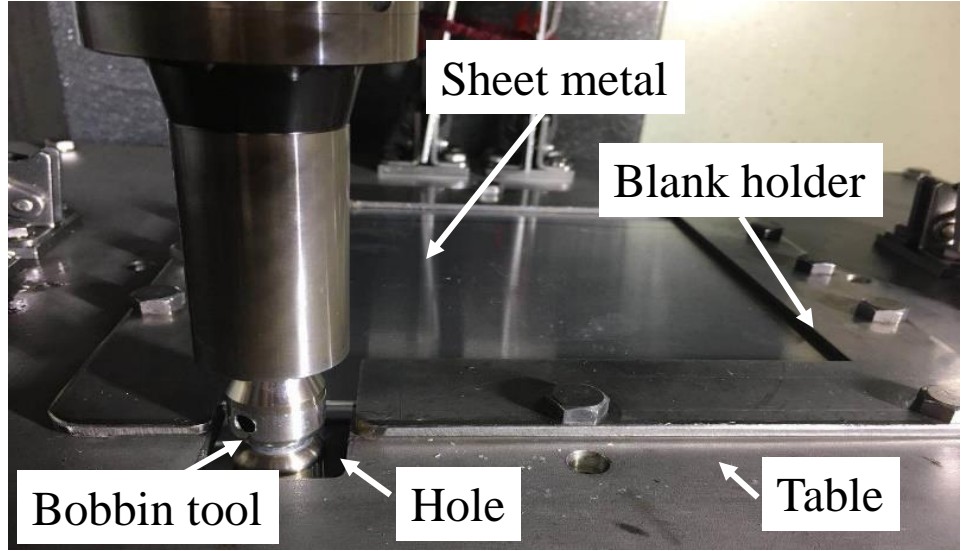

**Figure 2.** Appearance of the FSW setup.

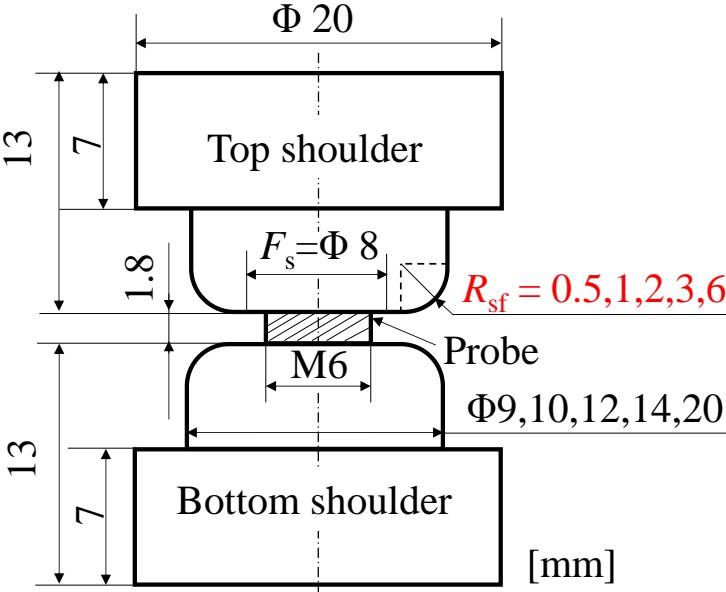

**Figure 3.** Schematic illustration of a bobbin tool.

**Table 1.** Tool size and welding parameters in BT-FSW experiments.

| | |
|---|---|
| Shoulder fillet radius $R_{sf}$ [mm] | 0.5, 1, 2, 3, 6 |
| Diameter of the flat area of shoulder $F_s$ [mm] | 8 |
| Shoulder diameter [mm] | 9, 10, 12, 14, 20 |
| Tool rotation direction | Clockwise |
| Tool rotation speed $N$ [rpm] | 1000–7000 |
| Tool welding speed $V$ [mm/min] | 1000–5000 |
| Gap between top and bottom shoulders $d_s$ [mm] | 1.8 |
| Processing type | Stir-in-plate |

Specimens with dimensions of 30 mm × 3 mm × 2 mm were used for cross-sectional observation, as shown in Figure 4a. Macroscopic images of the cross-section were taken using an image scanner (GT-X820, Seiko Epson Corp., Nagano, Japan), and the minimum thickness in each weld was measured using the images. The specimens were polished with #240 to #3000 sandpapers and $Al_2O_3$ suspensions with diameters of 1 μm and 0.3 μm. After polishing, the anodizing procedure was performed in a 2% aqueous $BH_3F$ solution at 30 V for 60 s at room temperature. The cross-sectional microstructure was observed using a metallurgical microscope (BX52M, Olympus Corp., Tokyo, Japan) with simple polarizing equipment and a digital imaging system. The grain size in the stir zone was evaluated from microscopic images using the line intercept method.

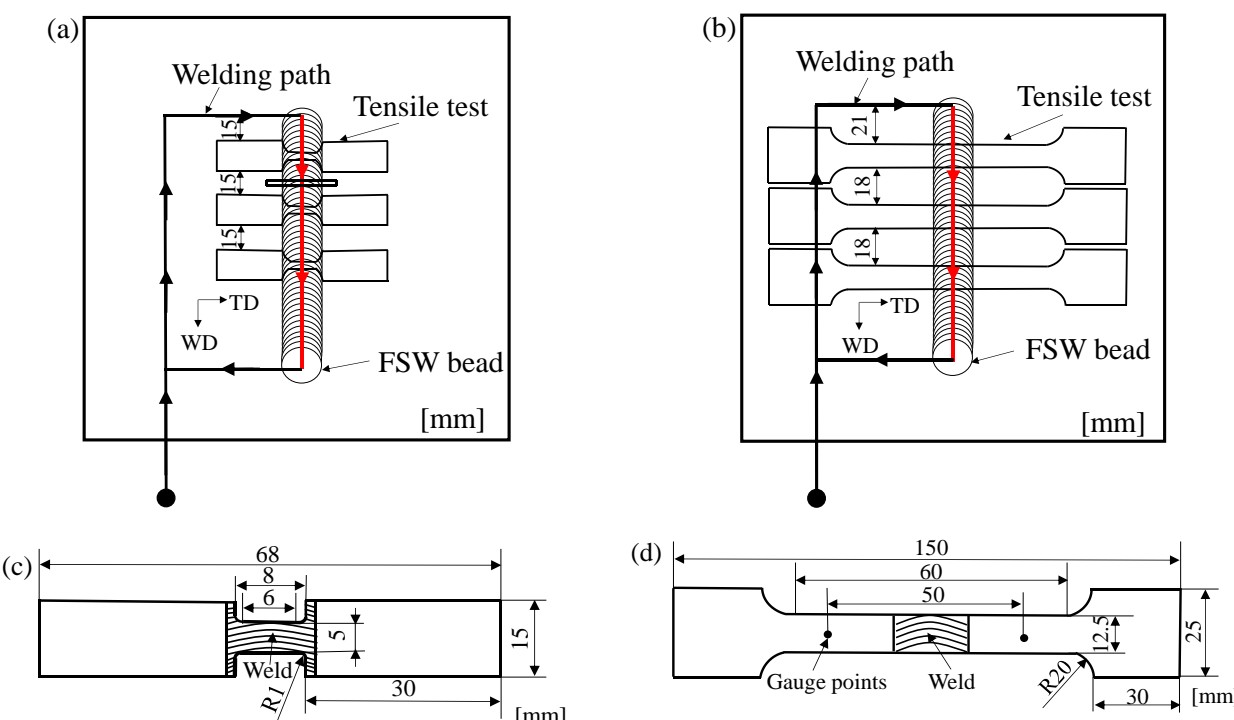

**Figure 4.** Schematic illustrations of the welding path and sampling location and size for (**a**) a within-weld tensile test and cross-sectional observation, (**b**) whole-weld tensile test, (**c**) a within-weld tensile test specimen, and (**d**) a whole-weld tensile test specimen.

Whole-weld and within-weld tensile specimens were prepared along the transverse direction of the welds, as shown in Figure 4a,b. The size of each specimen is shown in Figure 4c,d. The whole-weld tensile specimens were prepared according to Japanese Industrial Standard JIS Z 2241. The within-weld tensile specimens were uniquely designed so that the gauge length was completely covered by a stir zone. Tensile tests were performed using a universal testing machine (AGX-250kN, SHIMADZU CORP., Kyoto, Japan) at room temperature with a constant cross-head speed of 1 mm/min for the entire weld specimen and 0.2 mm/min for the stir zone. To comprehensively evaluate the effects of the distributions of the thickness and microstructure along the transverse direction on the tensile properties, the nominal stress was calculated assuming that the thickness was uniform over the base metal without flattening the specimen surface.

## 3. Results and Discussions

### 3.1. Defect Types and Process Windows

Figure 5 shows the process windows for (a) $R_{sf}$ = 0.5 mm, (b) $R_{sf}$ = 3 mm, and (c) $R_{sf}$ = 6 mm. The marks in the process windows were used for the classification of the

surface appearance of BT-FSW welds obtained in the present study. Figure 6 shows the typical appearances of each classification of welds.

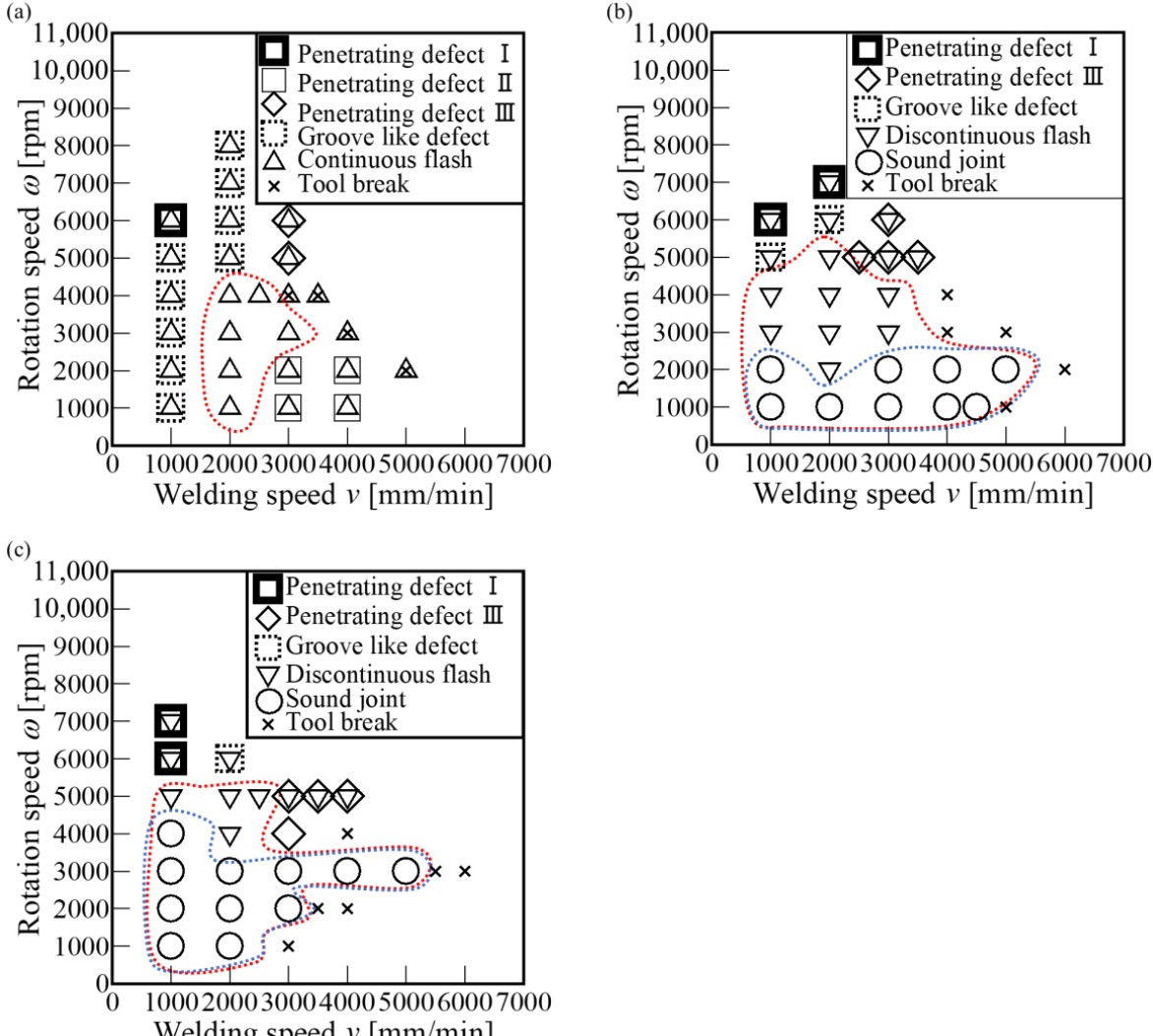

**Figure 5.** Process windows of shoulder fillet radii $R_{sf}$ of (**a**) 0.5 mm, (**b**) 3 mm, and (**c**) 6 mm (the red dotted line delineates the weldable area without welding defects, and the blue dotted delineates the sound joint area without welding defects or flash formation.

As shown in Figure 5a, in the $R_{sf}$ = 0.5 mm process window, penetrating defect I (bold line square) and the groove-like defect (break line square) occurred at relatively high rotation speeds and low welding speeds. Penetrating defect I was a continuous crack-like defect with a spiny end surface on the advancing side (AS) of the welds penetrating the thickness of the sheets, as shown in Figure 6b. The groove-like defect was generated on the advancing side (AS) of the backside or both sides of the welds, as shown in Figure 6e. In the case of conventional FSW, groove-like defects are generally generated owing to insufficient heat input, as reported by Kim et al. [32]. However, the groove-like defects in BT-FSW in the present study might have been caused by excessive heat input, which has a different generating principle than the groove-like defect observed in conventional FSW. Because penetrating defect I and the groove-like defects occurred at relatively high rotation speeds and low welding speeds and formed on the advancing side of the weld, they were considered identical defects; it was assumed that the groove-like defects extended and penetrated the sheet thickness to become penetrating defect I. Sued et al. reported similar types of defects as "cutting effects" [12]. In contrast, penetrating defect II (thin

square) occurred at a rotation speed lower than $N$ = 2000 rpm and a welding speed higher than $V$ = 3000 mm/min. Penetrating defect II was a continuous crack-like defect with a relatively smooth end surface that occurred on the advancing side of the welds penetrating the thickness of the sheets, as shown in Figure 6c. Penetrating defect II might have occurred due to the insufficient material flow owing to low heat input, that is, this type of defect might correspond to the groove-like defect in the conventional FSW [32]. Curiously, this type of defect formation due to insufficient heat input has rarely been focused on in BT-FSW research [10]. This is probably because such welding conditions were precluded as clearly failing conditions, in addition to being hidden by the occurrence of tool fracture, which will be subsequently described. Tool fracture (cross mark) occurred at low tool rotation speeds and high welding speeds because the load on the tool increased with decreasing heat input. The maximum welding speed of 5000 mm/min in this study is more than four times faster than that of any of the studies reviewed by Fuse et al. [10]. If the tool material has a higher strength, it is expected that the welding speed can be further improved. As shown in Figure 6d, at high rotation speed and high welding speeds (diagonal square), penetrating defect III, which was an intermittent welding defect characterized by crescent-shaped holes penetrating the sheet, was observed. Therefore, penetrating defect III was presumed to have been caused by the destabilization of the material flow as usually reported in conventional FSW [32]. Even in BT-FSW, unwelded joints due to the instability of material flow often become a problem [10]. In contrast to these welding defect conditions, weldable conditions were also observed; the weldable conditions, under which no welding defects were observed, were characterized by a rotation speed range of 1000 to 4000 rpm and welding speed range of 2000 to 3000 mm/min. However, because of the formation of a continuous pleated flash (triangle mark), the sound joint condition (circle mark) was not observed in the $R_{sf}$ = 0.5 mm process window. The pleated flash was formed on the retreading side (RS) and had a pleated shape and relatively large volume, as shown in Figure 6f. However, it can be inferred that the welding parameters had negligible influence on the formation of the pleated flash.

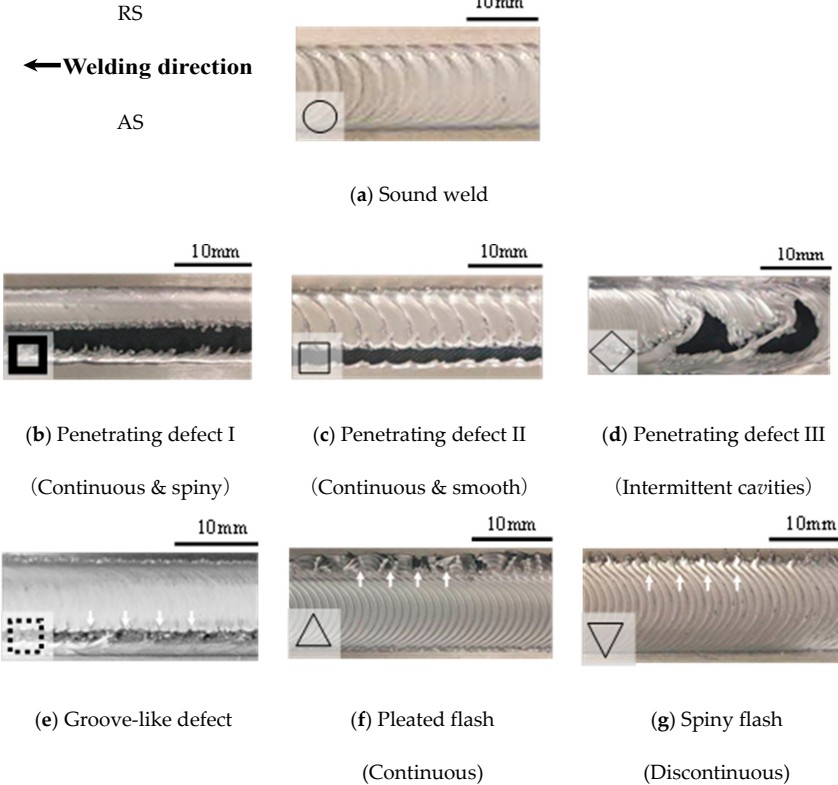

**Figure 6.** Appearances of the upper surfaces of the BT-FSW welds.

Figure 5b shows the $R_{sf}$ = 3 mm process window. When the $R_{sf}$ increased from 0.5 mm to 3 mm, penetrating defect I and groove-like defects were observed under high rotation speeds, penetrating defect II disappeared, and penetrating defect III remained almost unchanged. Consequently, the range of the weldable conditions increased significantly. Although the pleated flash was not observed at any welding parameter with the $R_{sf}$ = 3 mm tool, a discontinuous spiny flash (inverse triangle mark), which had a relatively small volume, was observed under high rotation-speed conditions, as shown in Figure 6g. This can be interpreted as an exposure of a spiny flash hidden by a pleated flash with a relatively large volume. Because the spiny flash was observed only under high rotation-speed conditions, sound joints without any flash or defect were observed under low rotation speeds in the weldable condition range, as shown in Figure 6a.

Figure 5c shows the $R_{sf}$ = 6 mm process window. Penetrating defect I, groove-like defects, and penetrating defect III were located in almost the same area as in the $R_{sf}$ = 3 mm process window. Penetrating defect II and pleated flash were not observed, similar to the $R_{sf}$ = 3 mm process window. In contrast, when the $R_{sf}$ was increased from 3 mm to 6 mm, the tool fracture range shifted towards the low-welding-speed side under the low rotation speed conditions. As a result, the weldable condition range became narrower. However, the sound joint condition range increased, and spiny flash conditions retracted to the high-rotation-speed side.

Although the continuous pleat-shaped flash was observed for all welding parameters for $R_{sf}$ = 0.5 mm, as shown in Figure 5a, the pleat-shaped flash disappeared for $R_{sf}$ values of 3 mm and 6 mm. Therefore, it can be inferred that the pleated flash was primarily caused by the shape of the shoulder. Moreover, relatively large volumes of pleated flash promote the formation of welding defects owing to a material shortage at the weld. Therefore, the weldable condition range becomes significantly narrower, as was observed in the case of $R_{sf}$ = 0.5 mm. Such a classification of flash morphology and quantitative effect shoulder fillet radius have not been reported in previous studies [10].

### 3.2. Average Flash Height and Weld Thickness

To investigate the effects of the shoulder fillet radius in more detail, the welding conditions of $N$ = 2000 rpm and $V$ = 2000 mm/min, which are weldable conditions over the $R_{sf}$ range of 0.5 mm to 6.0 mm, were selected from the process windows, as shown in Figure 5. Images of the weld surface and cross-sectional images of welds for different shoulder fillet radii, $R_{sf}$, of 0.5 mm, 1 mm, 2 mm, 3 mm, and 6 mm, are shown in Figure 7. The minimum thickness of the weld and the average height of the flash formation were measured from the cross-sectional images, as shown in Figure 8. The welds with $R_{sf}$ values of 0.5 mm, 1 mm, and 2 mm exhibited higher flash heights than the welds with $R_{sf}$ values of 3 mm and 6 mm. Thus, it can be concluded that a larger amount of material was ejected from the weld for $R_{sf}$ values of 0.5 mm, 1 mm, and 2 mm. The average flash height decreased from 1.4 mm to 0.2 mm as the $R_{sf}$ increased. As shown in Figure 8, except at the $R_{sf}$ value of 0.5 mm, the minimum weld thickness increased as the $R_{sf}$ increased from 0.5 mm to 6 mm; the weld thickness decreased from 1 mm to 6 mm. Because the weld obtained under the current welding conditions was relatively uniform along the welding direction, it can be inferred that the total amount of material in the cross-section was preserved before and after the welding. Therefore, the reduction amount of the material around the center of the weld was the same as the amount of material ejected as the flash. These above results clearly demonstrate that a relatively large shoulder edge fillet effectively prevent a flash formation accompanying weld thickness reduction, as researchers have believed [12–16,27].

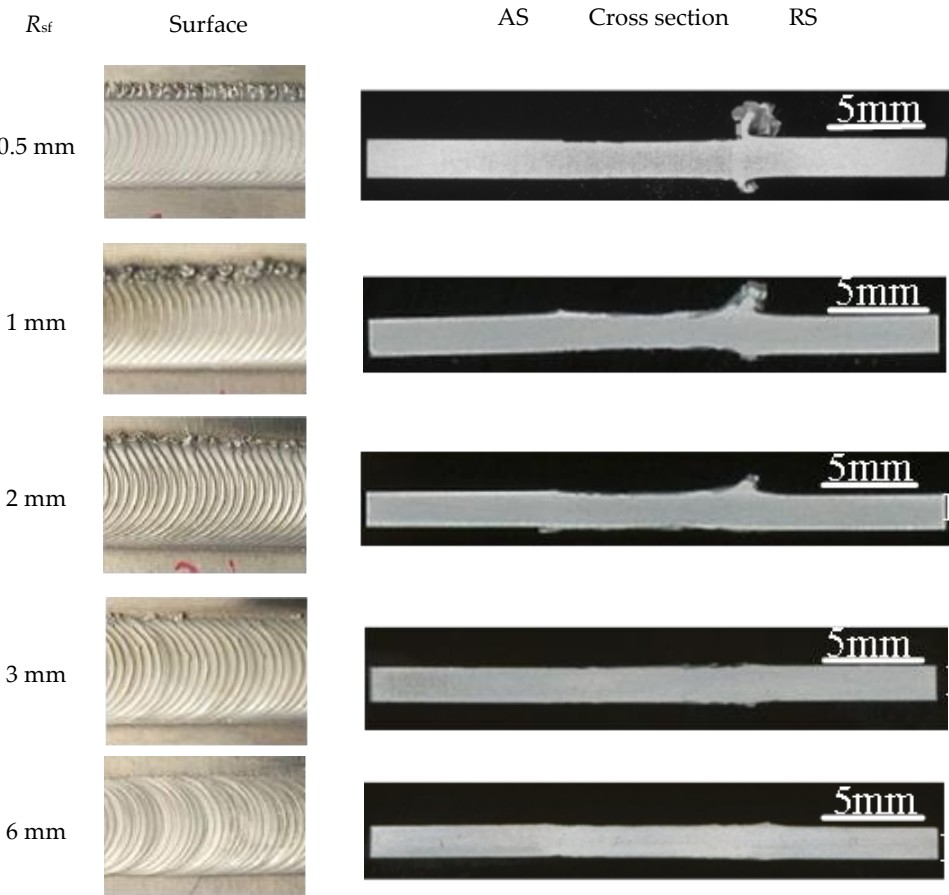

**Figure 7.** Appearances of weld surface and cross-sectional images of welds with different shoulder fillet radii $R_{sf}$ ($N$ = 2000 rpm, $V$ = 2000 mm/min, $F_s$ = 8 mm, $d_s$ = 1.8 mm).

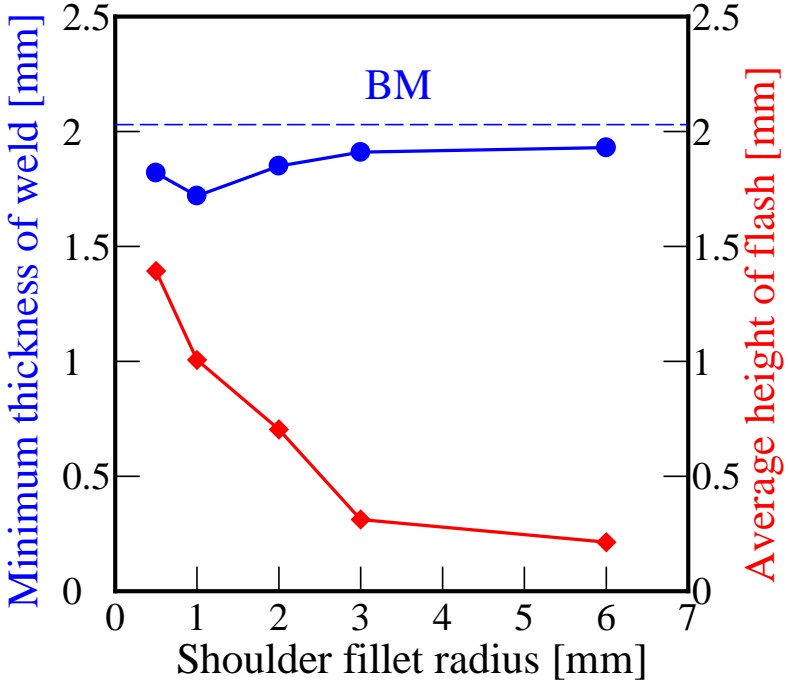

**Figure 8.** Relation between minimum thickness, average height of flash, and the shoulder fillet radii $R_{sf}$ ($N$ = 2000 rpm, $V$ = 2000 mm/min, $F_s$ = 8 mm, $d_s$ = 1.8 mm).

The typical cross-sectional microstructures with different shoulder fillet radii are shown in Figure 9. From Figure 9a, it can be observed that the pleated flash had a long cross-sectional length but a small cross-sectional area; therefore, the volume of the pleated flash was relatively small, even though its height was significantly high. Therefore, the trends of the average flash height and the minimum thickness of the weld were not in agreement. In addition, as shown in Figure 9a, a pleated flash was formed around the boundary between the SZ and TMAZ and had a continuous microstructure with the SZ on the inside and TMAZ on the outside. Based on the morphology and microstructural characteristics, it can be concluded that the pleated flashes were formed owing to the separation of the surface layer of the base metal due to stress concentration at the shoulder edge in front of the tool [27], similar to chip formation during micro-cutting. Therefore, the formation of pleated flash was suppressed with the increase of $R_{sf}$ due to reduction of stress concentration. Additionally, Liu et al. reported that, during hot micro-cutting, the growth of chips is slightly suppressed owing to the folding back of the material flow that forms the chips as the edge radius increases, and the effect of the material temperature on this tendency is small [33]. That is, when the $R_{sf}$ value is large, even if pleated flashes are generated, they are difficult to extrude in the out-of-plate direction.

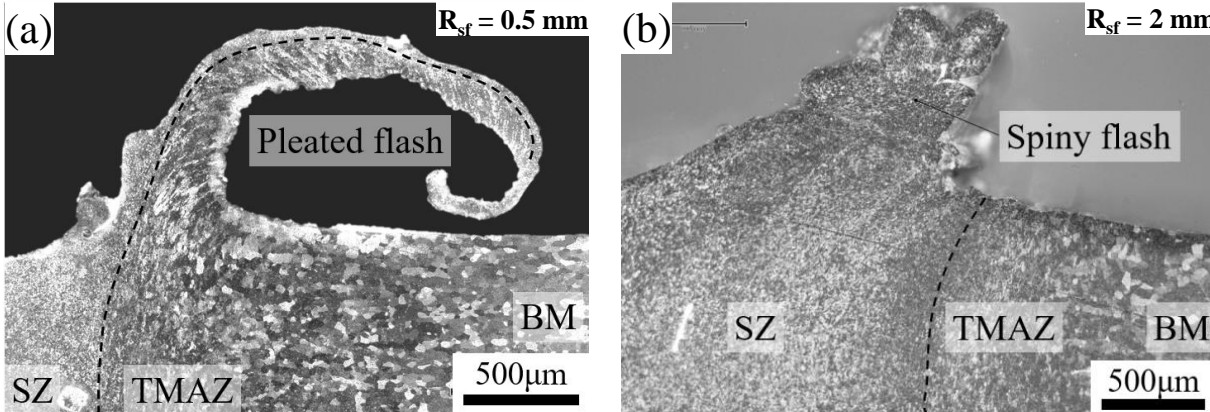

**Figure 9.** Cross-sectional microstructure of flash formed in the RS edge of the stir zones welded with the shoulder fillet radius $R_{sf}$ of (**a**) 0.5 mm and (**b**) 2 mm. ($N$ = 2000 rpm, $V$ = 2000 mm/min, $F_s$ = 8 mm, $d_s$ = 1.8 mm).

On the other hand, as shown in Figure 9b, the spiny flash had a microstructure continuous with only the SZ. Therefore, it was concluded that the spiny flash was formed by the extruding part of the SZ around the shoulder edge due to decreasing viscosity with excessively elevated temperature [10]. This conclusion agrees with the formation of spiny flash under high-heat-input conditions, as shown in Figure 5. As reported by Fuse and Badheka, a large diameter of the bobbin tool shoulder results in more flash formation owing to high heat generation, which facilitates the flow of plasticized material [34]. Note that, in this study, the large shoulder fillet radius produced less flash even though it resulted in a large contact area and heat generation. The reason why the spiny flash was suppressed as the $R_{sf}$ increased is assumed that the spiny flash generated due to the increase in heat input accompanying the increase in $R_{sf}$ were also folded towards the in-plate direction by the shoulder edge with the larger $R_{sf}$ [33], just like the pleated flash. However, the spiny flash was not observed for $R_{sf}$ values resulting in pleated flash formation. Considering that both flashes were located in the RS of SZ, it can be inferred that an early pleated flash formation suppressed the spiny flash formation.

### 3.3. Grain Size

Figure 10 shows the OM cross-sectional images of the base metal (BM) center of the stir zone (SZ). The SZs had finer equiaxial crystal grains than the BM. The average grain

sizes of the BM and SZs are shown in Figure 11. From the result, the average grain size in the SZs increased from 5.6 μm at $R_{sf}$ = 0.5 mm to 16.1 μm at $R_{sf}$ = 6 mm. The increase in grain size was caused by the increase in heat input as the contact diameter between the shoulder and sheet material increased.

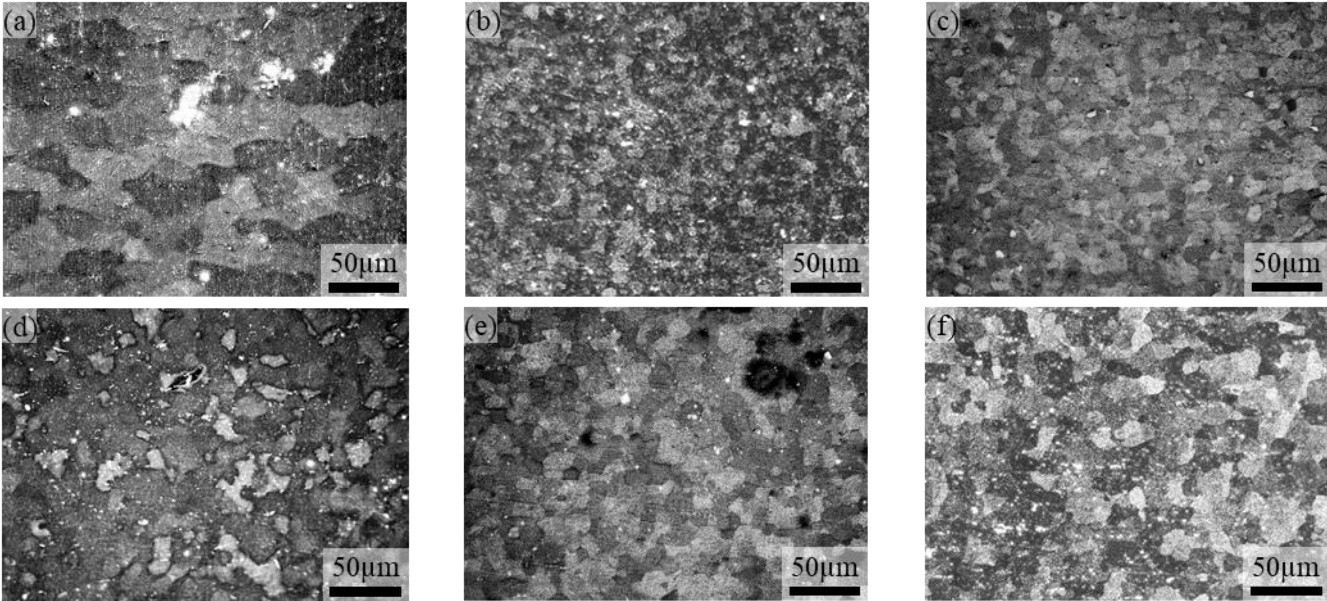

**Figure 10.** Optical microscope images of (**a**) base metal and stir zones with different shoulder fillet radii $R_{sf}$ of (**b**) 0.5 mm, (**c**) 1 mm, (**d**) 2 mm, (**e**) 3 mm, and (**f**) 6 mm, respectively. ($N$ = 2000 rpm, $V$ = 2000 mm/min, $F_s$ = 8 mm, tool gap = 1.8 mm).

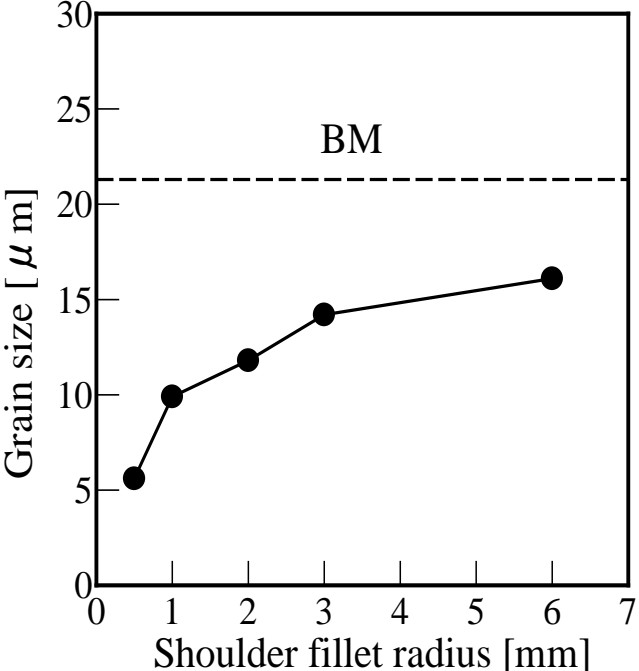

**Figure 11.** Relation between the grain size of the stir zone and the shoulder fillet radius $R_{sf}$ ($N$ = 2000 rpm, $V$ = 2000 mm/min, $F_s$ = 8 mm, $d_s$ = 1.8 mm).

For evaluating the relationship between FSW welding conditions and heat input, the following heat-input equation proposed by Frigaard et al. [35] is generally used:

$$q = \frac{4}{3}\pi^2 \mu PN(\phi/2)^3,$$  (1)

where $q$ is the frictional heat generated on the shoulder, $\mu$ is the friction coefficient between the shoulder and the material, $P$ is the pressure on the shoulder surface, $N$ is the rotation speed of the tool, and $\Phi$ is the shoulder diameter. Considering Equation (1), the heat input per unit welding length $Q$ is expressed by the following equation [36]:

$$Q = \frac{\alpha q}{V},$$

$$Q = \frac{\pi^2 \mu PN \phi^3}{6V},$$  (2)

where $\alpha$ is the heat input efficiency, and $V$ is the tool welding speed. Therefore, when $\alpha$, $\mu$, and $P$ are assumed to be constant, the following equation is obtained:

$$Q \propto \frac{N\phi^3}{V}$$  (3)

when the tool shoulders with different $R_{sf}$ values are plunged into the sheet with the same tool-plunging depth, the shoulder contacting diameter $\Phi$ in Equation (3) increases as $R_{sf}$ increases, resulting in more heat input per unit length $Q$. Therefore, in this study, the grain size increased from 5.6 μm to 16.1 μm with increasing heat input.

*3.4. Tensile Properties*

3.4.1. Whole Joint

Figure 12 and Table 2 shows the ultimate tensile strength (UTS), elongation of the entire joint (EWJ), and elongation within the weld (EWW). In this study, nominal elongations, which were calculated as the difference in the gauge length before and after the tensile test divided by the original gauge length, were used as the EWJ. The changes in the width of the weld during the test were divided by the original width of the weld to obtain the EWW. The error bars indicate standard deviation. The UTS of the joint was almost the same as or slightly higher than that of the BM; this is represented by the blue dotted line. All the joint tensile test specimens exhibited fractures in the BM. Thus, it can be concluded that the tensile strength of the weld was higher than that of the BM, even though the weld thickness was reduced. In contrast, the EWJ of the weld was smaller than that of the BM. The EWJ increased with increasing shoulder fillet radius from 0.5 mm to 1 mm, decreased from 1 mm to 3 mm, and remained constant from 3 mm to 6 mm. The EWJ trend was similar to the EWW trend. Figure 13 shows the appearance of the welds after the tensile test. As shown in Figure 13, the weld with $R_{sf} = 1$ mm, which had the lowest minimum thickness, was constricted in the welding direction, which is perpendicular to the tensile direction. Thus, the weld with $R_{sf} = 1$ mm yielded a lower load than the BM owing to the reduction in the thickness in the SZ. However, because the tensile strength of the weld was greater than that of the BM, it can be assumed that the deterioration of the tensile strength due to thickness reduction was compensated by the large strain hardening of the microstructure of the SZ. In contrast, although the weld thickness decreased for all the welds, the yield strength of the welds was higher than that of the BM; thus, it can be concluded less elongation occurred within the weld.

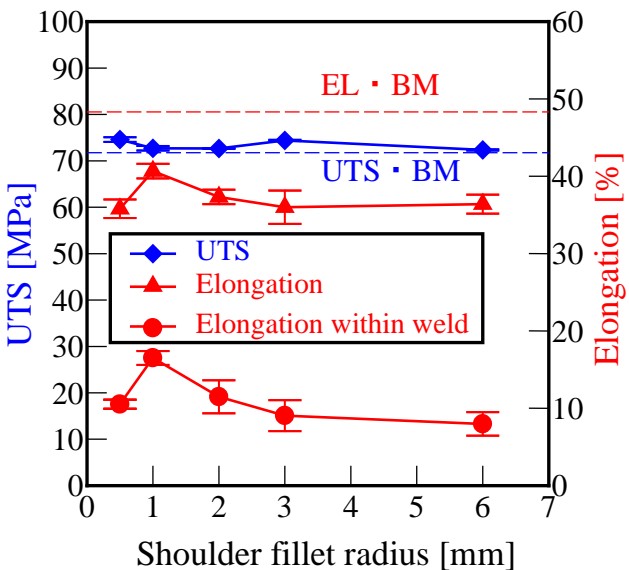

**Figure 12.** Relation between shoulder fillet radii $R_{sf}$ and tensile properties of whole-weld specimens ($N$ = 2000 rpm, $V$ = 2000 mm/min, $F_s$ = 8 mm, $d_s$ = 1.8 mm).

**Table 2.** Tensile test results for whole-weld specimens.

| Sample | UTS [MPa] | Joint Efficiency [%] | EWJ [%] | EWW [%] | Fracture Location |
|---|---|---|---|---|---|
| Base metal | 71.8 | - | 48.3 | - | - |
| $R_{sf}$ = 0.5 mm | 74.6 ± 0.5 | 103.9 | 35.8 ± 1.2 | 10.5 ± 0.6 | Base metal |
| $R_{sf}$ = 1 mm | 72.7 ± 0.5 | 101.3 | 40.7 ± 0.9 | 16.5 ± 0.9 | Base metal |
| $R_{sf}$ = 2 mm | 72.7 ± 0.1 | 101.3 | 37.3 ± 0.9 | 11.5 ± 2.1 | Base metal |
| $R_{sf}$ = 3 mm | 74.4 ± 0.1 | 103.6 | 36.0 ± 2.2 | 9.1 ± 2.0 | Base metal |
| $R_{sf}$ = 6 mm | 72.4 ± 0.1 | 100.8 | 36.4 ± 1.2 | 8.0 ± 1.5 | Base metal |

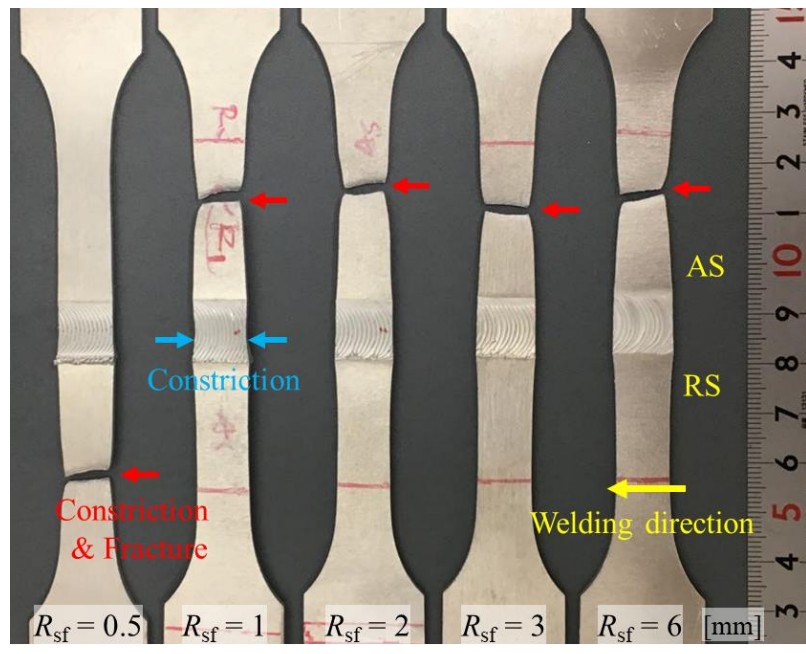

**Figure 13.** Appearance of whole-joint tensile specimens with different shoulder fillet radii $R_{sf}$.

### 3.4.2. Within Weld

The UTS and elongation values of the SZ tensile specimens are shown in Figure 14 and Table 3. All of the specimens fractured near the center of the SZ. The UTS of the SZ was relatively higher than that of the BM, as represented by the blue dotted line, owing to the refinement of the grains in the SZ. The UTS was the lowest for $R_{sf}$ = 1 mm and gradually increased with increasing $R_{sf}$ from 1 mm to 3 mm. However, the UTS of $R_{sf}$ = 3 mm was slightly larger than that of $R_{sf}$ = 6 mm. The tendency of elongation was similar to that of the UTS, and all elongations in the SZ were lower than that in the BM. As shown in Figure 8, the average grain sizes in the SZs increase monotonically with increasing $R_{sf}$. That is, the changing tendency of the UTS and elongation was similar to that of the minimum weld thickness shown in Figure 8 rather than the tendency of grain size. It was concluded that the minimum weld thickness had a more significant effect on the UTS and elongation than the grain size. However, based on the Hall–Petch relation, the UTS is also affected by the grain size. The effect of grain refinement became apparent in each comparison of the UTS between $R_{sf}$ = 3 mm and 6 mm and between $R_{sf}$ = 0.5 mm and 2 mm; considering these comparisons, it can be inferred that the changing tendency of UTS differed from that of the minimum thickness, as shown in Figure 8.

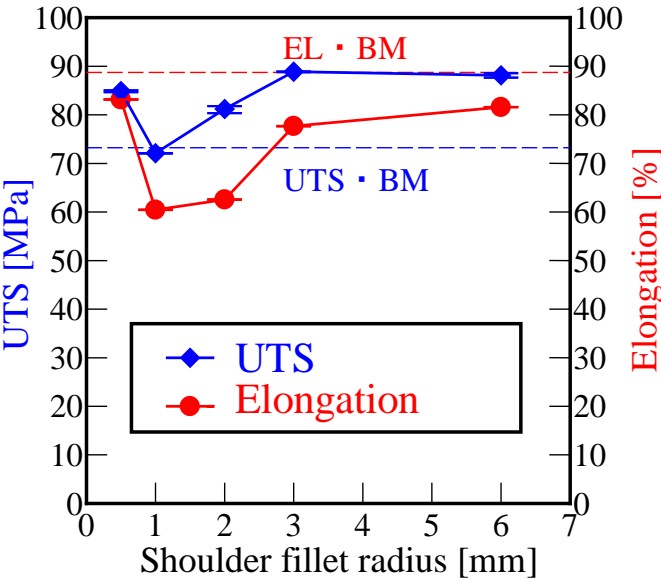

**Figure 14.** Relation between shoulder fillet radii $R_{sf}$ and tensile properties of within-weld specimens ($N$ = 2000 rpm, $V$ = 2000 mm/min, $F_s$ = 8 mm, $d_s$ = 1.8 mm).

**Table 3.** Tensile test results for within-weld specimens.

| Sample | UTS [MPa] | Elongation [%] |
|---|---|---|
| Base metal | 73.2 | 88.7 |
| $R_{sf}$ = 0.5 mm | 84.9 ± 0.2 | 83.2 ± 0.0 |
| $R_{sf}$ = 1 mm | 72.1 ± 0.0 | 60.4 ± 0.1 |
| $R_{sf}$ = 2 mm | 81.2 ± 0.7 | 62.6 ± 0.0 |
| $R_{sf}$ = 3 mm | 88.9 ± 0.1 | 77.7 ± 0.1 |
| $R_{sf}$ = 6 mm | 88.1 ± 0.5 | 81.6 ± 0.0 |

## 4. Conclusions

In this study, BT-FSW tools with five different shoulder fillet radii of 0.5 mm, 1 mm, 2 mm, 3 mm, and 6 mm were used to elucidate the effect of the shoulder fillet radius on the formation of welding defects, flash formation, weld thickness reduction, tensile property, and grain size. A1050-O sheets were used for the experiments, and the following conclusions were obtained.

1. The average height of flash decreased with increasing shoulder fillet radius from 0.5 mm to 6 mm. Therefore, it can be concluded that increasing the shoulder fillet radius is an effective approach to decrease flash formation.
2. The shoulder fillet radius had an obvious effect on the flash shape under the same welding conditions. Continuous flash formation with a large volume was observed for a shoulder fillet radius of 0.5 mm. In contrast, discontinuous flash formation with a small volume was observed under relatively high heat-input welding conditions for shoulder fillet radii of 3 mm and 6 mm.
3. The formation of welding defects decreased under both relatively high and low heat-input welding conditions with increasing shoulder fillet radius from 0.5 mm to 3 mm and 6 mm. Additionally, a sound weld area without any flash or welding defect formation was observed when the radius was increased from 0.5 mm to 3 mm and 6 mm.
4. The minimum weld thickness decreased when the shoulder fillet radius changed from 0.5 mm to 1 mm, then increased when the radius increased from 1 mm to 3 mm, and finally became almost constant above a radius of 3 mm. The changing tendency of the UTS was similar to that of the minimum thickness.
5. The grain size of the stir zone increased with increasing shoulder fillet radius, which can be attributed to the increase in the heat input with a larger shoulder fillet radius.
6. From the viewpoint of suppressing flash formation and tool fracture at high welding speeds, the optimum shoulder fillet radius is 3 mm.

**Author Contributions:** Conceptualization, T.M., W.J. and M.O. (Masaaki Otsu); Methodology, H.M., T.M., W.J., M.O. (Masato Okada) and M.O. (Masaaki Otsu); Validation, W.J. and M.O. (Masato Okada); Investigation, H.M.; Data curation, H.M.; Writing—original draft, H.M.; Writing—review & editing, T.M., W.J., M.O. (Masato Okada) and M.O. (Masaaki Otsu); Supervision, T.M., M.O. (Masato Okada) and M.O. (Masaaki Otsu); Project administration, M.O. (Masaaki Otsu); Funding acquisition, M.O. (Masaaki Otsu). All authors have read and agreed to the published version of the manuscript.

**Funding:** This research received no external funding.

**Conflicts of Interest:** The authors declare no conflict of interest.

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
