# Peer review of "Effect of Shoulder Fillet Radius on Welds in Bobbin Tool Friction Stir Welding of A1050"

_metals, doi:10.3390/met12111993_

Round 1

Reviewer 1 Report

See attached document for comments.

Author Response

Thank you for giving us the opportunity to revise our manuscript submitted to the metals. We carefully referred all reviewer comments and made revision in our manuscript. Thanks to the precise comments from reviewers, we were able to clarify the novelty of present research. Please review the following our responses to reviewer comments along with the revised manuscript. All revised parts are highlighted in yellow.

(Reviewer 1 Comment 1) Perform a thorough literature review of FSW bobbin tools. There are many such studies. A quick search via the internet will yield many bobbin tool research papers. I would recommend a target range of 10-12 papers regarding FSW bobbin tools be added.

  • Thank you for your advice. Certainly, we were missing a lot of references to the literature on BT-FSW. We have added the literatures on BT-FSW shown below.
  1. Threadgill, P.L.; Ahmed, M.M.Z.; Martin, J.; Jonathan, G.P.; Wynne, B.P. The use of bobbin tools for friction stir welding of aluminium alloys. Sci. Forum. 2010, 638–642, 1179-1184.
  2. Fuse, K.; Badheka, V. Bobbin tool friction stir welding: a review, Technol. Weld. Join. 2018, 24, 277-304.
  3. Zhou, L.; Li, G.H.; Zha, G.D.; Shu, F.Y.; Liu, H.J.; Feng, J.C. Effect of rotation speed on microstructure and mechanical properties of bobbin tool friction stir welded AZ61 magnesium alloy, Technol. Weld. Join. 2018, 23, 1362-1718.
  4. Sued, M.K.; Pons, D.; Lavroff, J.; Wong, E.H. Design features for bobbin friction stir welding tools: Development of a conceptual model linking the underlying physics to the production process, Des. 2014, 54, 632-643.
  5. Wang, F.F.; Li, W.Y.; Shen, J.; Zhang, Z.H.; Li, J.L.; dos Santos, J.F. Global and local mechanical properties and microstructure of bobbin tool friction-stir-welded Al-Li alloy. Technol. Weld. Join. 2016, 21, 479–483.
  6. Zhou, L.; Li, G.H.; Liu, C.L.; Wang, J.; Huang, Y.X.; Feng, J.C.; Meng, F.X. Microstructural characteristics and mechanical properties of Al–Mg–Si alloy self-reacting friction stir welded joints. Technol. Weld. Join. 2017, 22, 438–445.
  7. Hou, J.C.; Liu, H.J.; Zhao, Y.Q.; Influences of rotation speed on microstructures and mechanical properties of 6061-T6 aluminum alloy joints fabricated by self-reacting friction stir welding tool. J. Adv. Manuf. Technol. 2014, 73, 1073–1079.
  8. Liu, H.J.; Hou, J.C.; Guo, H.; Effect of welding speed on microstructure and mechanical properties of selfreacting friction stir welded 6061-T6 aluminum alloy. Des. 2013, 50, 872–878.
  9. Wan, L.; Huang, Y.X.; Guo, W.Q.; Lv, S.X.; Feng. J.C., Mechanical properties and microstructure of 6082-T6 aluminum alloy joints by self-support friction stir welding. Mater. Sci. Technol. 2014, 30, 1243-1250.
  10. Chen, S.; Li, H.; Lu, S.; Ni, R.; Dong, J. Temperature measurement and control of bobbin tool friction stir welding. J. Adv. Manuf. Technol. 2016, 86, 337–346.
  11. Longhurst, W.R.; Cox, C.D.; Gibson, B.T.; Brian, T.; Cook, G.E.; Strauss, A.M.; Wilbur, I.C.; Osborne, B.E. Development of friction stir welding technologies for in-space manufacturing. J. Adv. Manuf. Technol. 2017, 90, 81–91.
  12. Zhou, L.; Li, G.H.; Liu, C.L.; Wang, J.; Huang, Y.X.; Feng, J.C.; Meng, F.X. Effect of rotation speed on microstructure and mechanical properties of self-reacting friction stir welded Al-Mg-Si alloy. J. Adv. Manuf. Technol. 2017, 89, 3509–3516.
  13. Li, Y.; Sun, D.; Gong, W.; Effect of tool rotational speed on the microstructure and mechanical properties of bobbin tool friction stir welded 6082-T6 aluminum alloy, metals 2019, 9, 894-905.

(Reviewer 1 Comment 2) State how this FSW bobbin tool research is different from the ones presented in the literature review (see comment above).

  • We have improved the comparison with references added in above response and the novelty of present research in the introduction part as following.

L44-59: In the conventional FSW, since the stirring at the bottom of the joint along the plate thickness direction is relatively weak, kissing bonds or root defects may occur due to insufficient stirring and heat input at the bottom [7,8]. Second, it is difficult to install a back support, making it difficult to apply to large hollow structural parts. In order to solve these problems, Bobbin Tool friction stir welding (BT-FSW) was developed as a derivative technology of FSW, in which the material is sandwiched with shoulders from both sides. In some literature, it is also called Self-Support FSW or Self-Reacting FSW [9-22]. A schematic illustration of BT-FSW, which is an FSW variant, is shown in Figure 1. The bobbin tool consists of a probe, top shoulder, and bottom shoulder. BT-FSW has various advantages, such as the omission of the backing bar and an absence of root defect formation. Furthermore, in BT-FSW, since the tool load can be canceled by the internal stress of the tool, the rigidity required for the device and fixture is low, and the heat input is relatively high, so the material tends to soften, so the welding speed can be increased. It is possible to improve the efficiency compared to the conventional FSW [9,10]. Zhou et al. explained that flash was formed due to heat accumulation at retreating side in magnesium alloy FSW using a bobbin tool with upper and lower shoulders of different diameters [11].

L72-82: Regarding shoulder geometry, although a simple shoulder shape, such as a flat shoulder, can be manufactured easily, a flat surface shoulder often leads to an excessive flash formation, as Unfried-Silgado et al. reported on AA1100 aluminum [24]. Casalino et al. also investigated the effects of shoulder geometry on weldability [25], and reported that a flat shoulder is sensitive to the process parameters and results in flash and welding defect formation. In contrast, a convex shoulder design can increase the contact area with the workpiece and facilitate the joining of workpieces of different thicknesses, as Nishihara and Nagasaka reported [24]. A concave shoulder design is commonly used to prevent material spilling and compress material around the probe during FSW, as demonstrated by Scialpi et al. [27] and reviewed by El-Moayed et al. [28].

L94-103: De Giorgi et al. reported that a flat shoulder with a fillet radius of 1 mm results in negligible flash formation [31]. Recently, Jiang et al. conducted a BT-FSW experiment using a bobbin tool with a relatively large fillet radius of 6 mm at the edge of the shoulder with a total plunging depth of 0.2 mm and reported that a sound joint was obtained, and almost no flash formation was observed [32]. Other literatures also use welding tools with relatively large radius fillets on the shoulder edges [12-16,27]. However, the effects of changes in shoulder fillet radius on the flash formation and welding defects have not been systematically evaluated yet. Therefore, it is difficult to obtain a practical process window for the BT-FSW that can be used as a guideline for selecting welding parameters and designing tool.

(Reviewer 1 Comment 3) State if the results agree with prior studies or do they differ?

  • We have improved the explanation about the comparison of our results along with the previous researches in the result and discussion part as follows.

L182-186: Because penetrating defect I and the groove-like defects occurred at relatively high rotation speeds and low welding speeds and formed on the advancing side of the weld, they were considered identical defects; it was assumed that the groove-like defects extended and penetrated the sheet thickness to become penetrating defect I. Sued et al. reported similar types of defects as “cutting effects” [12].

L191-196: Penetrating defect II might have occurred due to the insufficient material flow owing to low heat input, that is, this type of defect might correspond to the groove-like defect in the conventional FSW [33]. Curiously, this type of defect formation due to insufficient heat input has rarely been focused in BT-FSW research [10]. This is probably because such welding conditions were precluded as clearly failing conditions, in addition to be hidden by the occurrence of tool fracture, which will be described as following.

L198-201: The maximum welding speed of 5000 mm/min in this study is more than four times faster than that of any of the studies reviewed by Fuse et al. [10]. If the tool material has a higher strength, it is expected that the welding speed can be further improved.

L203-206: Therefore, penetrating defect III was presumed to have been caused by the destabilization of the material flow as usually reported in conventional FSW [33]. Even in BT-FSW, unwelded joints due to instability of material flow often become a problem [10].

L240-241: Such classification of flash morphology and quantitative effect shoulder fillet radius have not been reported in previous studies [10].

L260-262: These above results clearly demonstrate that a relatively large shoulder edge fillet effectively prevent a flash formation accompanying weld thickness reduction, as researchers have believed [12-16,27].

(Reviewer 1 Comment 4) In the conclusions, provide the optimum tool geometry and processing parameters (tool rotational speed and travel rate) found from this study that leads to increased weld quality.

  • We have added a discussion on optimal tool geometry to the conclusion. On the other hand, since we have not tried to optimize on the welding conditions in this study, We have avoided to mention the optimum welding conditions.

L404-405: 6. From the viewpoint of suppressing flash formation and tool fracture at high welding speeds, the optimum shoulder fillet radius is 3 mm.

Reviewer 2 Report

Review report on the topic ‘Effect of Shoulder Fillet Radius on Welds in Bobbin Tool Friction Stir Welding of A1050’. The work is presented well. The comments to improve the quality of the manuscript are listed below:

  1. Please omit the unnecessary information and add the key conclusion of the work at the end of the abstract section.
  2. Please add a separate section to discuss the novelty of the work.
  3. The introduction section is presented roughly. Add more references and try to make a bridge between current and previously published work. Refer to following: https://doi.org/10.1007/s11665-022-06832-2; https://doi.org/10.1007/s11665-022-06822-4.
  4. Add more discuss about selection of tool material selection and their profile.
  5. Mention the standard used for testing sample preparation.
  6. Discuss the mechanism of the pleated and spinly flash formation.
  7. Provide good quality optical image. It is difficult to reach any conclusion related to grain size.
  8. Add the reference for each equation.
  9. Add the image of the fractured tensile specimen and also tensile properties in separate section along with fracture location and joint efficiency: https://doi.org/10.1007/s11661-020-05660-0.
  10. Try to relate the tensile properties with microstructure evolution.

Overall work is good and can be accepted after following minor corrections.

Author Response

Dear, Professors

Thank you for giving us the opportunity to revise our manuscript “Effect of Shoulder Fillet Radius on Welds in Bobbin Tool Fric-tion Stir Welding of A1050” submitted to the metals. We carefully referred all reviewer comments and made revision in our manuscript. Thanks to the precise comments from reviewers, we were able to clarify the novelty of present research. Please review the following our responses to reviewer comments along with the revised manuscript. All revised parts are highlighted in yellow. 
Sincerely, 
Takuya MIURA
Joining and Welding Research Institute, Osaka University 
11-1 Mihogaoka, Ibaraki, Osaka 567-0047
Tel./Fax: +81-6-6879-8663
Email: miura@jwri.osaka-u.ac.jp

(Reviewer 2 Comment 1) Please omit the unnecessary information and add the key conclusion of the work at the end of the abstract section.
    We have added the critical information to abstract. On the other hand, we also considered reducing abstract, but decided that it was useful information for our readers and did not reduce it. 

L8-13: Abstract: In this study, five bobbin tools with different shoulder fillet radii were employed for the bobbin tool friction stir welding (BT-FSW) of A1050-O sheets to systematically evaluate the effects of shoulder fillet radius on the welding defect formation, flash formation, weld thickness, grain size of the stir zone, and tensile properties. The quality classifications of joints appearance were summarized as process windows, and the appropriate welding condition range for each shoulder fillet radius were clarified.

(Reviewer 2 Comment 2) Please add a separate section to discuss the novelty of the work.
    Thank you very much for your helpful advice. We have added a point-by-point explanation of the novelty of this research and comparison along with previous studies. On the other hand, we considered adding a separate part about the novelty, but omitted it because it would be redundant.

L182-186: Because penetrating defect I and the groove-like defects occurred at relatively high rotation speeds and low welding speeds and formed on the advancing side of the weld, they were considered identical defects; it was assumed that the groove-like defects extended and penetrated the sheet thickness to become penetrating defect I. Sued et al. reported similar types of defects as “cutting effects” [12].
L191-196: Penetrating defect II might have occurred due to the insufficient material flow owing to low heat input, that is, this type of defect might correspond to the groove-like defect in the conventional FSW [33]. Curiously, this type of defect formation due to insufficient heat input has rarely been focused in BT-FSW research [10]. This is probably because such welding conditions were precluded as clearly failing conditions, in addition to be hidden by the occurrence of tool fracture, which will be described as following.
L198-201: The maximum welding speed of 5000 mm/min in this study is more than four times faster than that of any of the studies reviewed by Fuse et al. [10]. If the tool material has a higher strength, it is expected that the welding speed can be further improved.
L203-206: Therefore, penetrating defect III was presumed to have been caused by the destabilization of the material flow as usually reported in conventional FSW [33]. Even in BT-FSW, unwelded joints due to instability of material flow often become a problem [10].
L240-241: Such classification of flash morphology and quantitative effect shoulder fillet radius have not been reported in previous studies [10].
L260-262: These above results clearly demonstrate that a relatively large shoulder edge fillet effectively prevent a flash formation accompanying weld thickness reduction, as researchers have believed [12-16,27].

(Reviewer 2 Comment 3) The introduction section is presented roughly. Add more references and try to make a bridge between current and previously published work. Refer to following: https://doi.org/10.1007/s11665-022-06832-2; https://doi.org/10.1007/s11665-022-06822-4.
    Thank you very much for your helpful advice and suggestion. We have improved the introduction about comparison with previous researches and the novelty of present research as following. 

L44-59: In the conventional FSW, since the stirring at the bottom of the joint along the plate thickness direction is relatively weak, kissing bonds or root defects may occur due to insufficient stirring and heat input at the bottom [7,8]. Second, it is difficult to install a back support, making it difficult to apply to large hollow structural parts. In order to solve these problems, Bobbin Tool friction stir welding (BT-FSW) was developed as a derivative technology of FSW, in which the material is sandwiched with shoulders from both sides. In some literature, it is also called Self-Support FSW or Self-Reacting FSW [9-22]. A schematic illustration of BT-FSW, which is an FSW variant, is shown in Figure 1. The bobbin tool consists of a probe, top shoulder, and bottom shoulder. BT-FSW has various advantages, such as the omission of the backing bar and an absence of root defect formation. Furthermore, in BT-FSW, since the tool load can be canceled by the internal stress of the tool, the rigidity required for the device and fixture is low, and the heat input is relatively high, so the material tends to soften, so the welding speed can be increased. It is possible to improve the efficiency compared to the conventional FSW [9,10]. Zhou et al. explained that flash was formed due to heat accumulation at retreating side in magnesium alloy FSW using a bobbin tool with upper and lower shoulders of different diameters [11].
L72-82: Regarding shoulder geometry, although a simple shoulder shape, such as a flat shoulder, can be manufactured easily, a flat surface shoulder often leads to an excessive flash formation, as Unfried-Silgado et al. reported on AA1100 aluminum [24]. Casalino et al. also investigated the effects of shoulder geometry on weldability [25], and reported that a flat shoulder is sensitive to the process parameters and results in flash and welding defect formation. In contrast, a convex shoulder design can increase the contact area with the workpiece and facilitate the joining of workpieces of different thicknesses, as Nishihara and Nagasaka reported [24]. A concave shoulder design is commonly used to prevent material spilling and compress material around the probe during FSW, as demonstrated by Scialpi et al. [27] and reviewed by El-Moayed et al. [28]. 

L94-103: De Giorgi et al. reported that a flat shoulder with a fillet radius of 1 mm results in negligible flash formation [31]. Recently, Jiang et al. conducted a BT-FSW experiment using a bobbin tool with a relatively large fillet radius of 6 mm at the edge of the shoulder with a total plunging depth of 0.2 mm and reported that a sound joint was obtained, and almost no flash formation was observed [32]. Other literatures also use welding tools with relatively large radius fillets on the shoulder edges [12-16,27]. However, the effects of changes in shoulder fillet radius on the flash formation and welding defects have not been systematically evaluated yet. Therefore, it is difficult to obtain a practical process window for the BT-FSW that can be used as a guideline for selecting welding parameters and designing tool.

(Reviewer 2 Comment 4) Add more discuss about selection of tool material selection and their profile.
    We have improved the explanation about the tool geometry and materials as following. 

L118-127: Five bobbin tools with shoulder fillet radii (Rsf) of 0.5, 1, 2, 3, and 6 mm were prepared. A tool with a shoulder fillet radius of 1 mm is often used as a filleted tool [31]. The shoulders were simply flat to clarify the influence of the shoulder fillet radius. The diameter of the flat plane of the shoulder (Fs) was kept constant at Φ8 mm for all tools, which means that the total diameter of the shoulders varied between Φ9 mm, Φ10 mm, Φ12 mm, Φ14 mm, and Φ20 mm with changes in Rsf. The tools were made of stainless steel (JIS SUS304), which is easily available and has a good balance of high-temperature strength and workability. M6 right-hand screw bolts were used to connect the top and bottom shoulders, and the gaps between the shoulders were adjusted to a constant value of 1.8 mm, which is 0.2 mm smaller than the sheet thickness, using a feeler gauge.

(Reviewer 2 Comment 5) Mention the standard used for testing sample preparation.
    We have added the detailed explanation about the design of tensile specimens. 

L151-153: The whole-weld tensile specimens were prepared according to Japanese Industrial Standards JIS Z 2241. The within-weld tensile specimens were uniquely designed so that the gauge length was completely covered by a stir zone.

(Reviewer 2 Comment 6) Discuss the mechanism of the pleated and spinly flash formation.
    We have improved the discussion about flash formation as following. Thank you for giving us the opportunity to improve.

L275-284: Based on the morphology and microstructural characteristics, it can be concluded that the pleated flashes were formed owing to the separation of the surface layer of the base metal due to stress concentration at the shoulder edge in front of the tool [27], similar to chip formation during micro-cutting. Therefore, the formation of pleated flash was suppressed with increase of Rsf due to reduction of stress concentration. Additionally, Liu et al. reported that during hot micro-cutting, the growth of chips is slightly suppressed owing to the folding back of the material flow that forms the chips as the edge radius increases and, the effect of material temperature on this tendency is small [34]. That is, when Rsf value is large, even if pleated flashes are generated, they are difficult to be extruded to the out-of-plate direction.
L289-291: Therefore, it was concluded that the spiny flash was formed by the extruding part of the SZ around the shoulder edge due to lowering viscosity with excessively elevated temperature [10].
L296-L299: The reason why the spiny flash was suppressed as the Rsf increased is assumed that the spiny flash generated due to the increase in heat input accompanying the increase in Rsf were also folded towards the in-plate direction by the shoulder edge with the larger Rsf [34], just like the pleated flash.

(Reviewer 2 Comment 7) Provide good quality optical image. It is difficult to reach any conclusion related to grain size.
    We apologize for the inconvenience. We have improved the quality of optical micrographs of Figure 10.

(Reviewer 2 Comment 8) Add the reference for each equation.
    We added reference of [**] for equation (2) in L322.

(Reviewer 2 Comment 9) Add the image of the fractured tensile specimen and also tensile properties in separate section along with fracture location and joint efficiency: https://doi.org/10.1007/s11661-020-05660-0.
    We have improved Figure 13 and added Tables 2 and 3. At the same time, Figure 14 was wrong, so we have replaced Figure 14. 

(Reviewer 2 Comment 10) Try to relate the tensile properties with microstructure evolution.
    We have improved the explanation about relationship between the grain sizes and tensile properties as following. 
L365-374: As shown in Figure 8, the average grain sizes in the SZs increase monotonically with increasing Rsf. That is, the changing tendency of the UTS and elongation was similar to that of the minimum weld thickness shown in Figure 8 rather than the tendency of grain size. It was concluded that the minimum weld thickness had a more significant effect on the UTS and elongation than the grain size. However, based on the Hall-Petch relation, the UTS is also affected by the grain size. The effect of grain refinement became apparent in each comparison of the UTS between Rsf = 3 mm and 6 mm and between Rsf = 0.5 mm and 2 mm; considering these comparisons, it can be inferred that the changing tendency of UTS differed from that of the minimum thickness, as shown in Figure 8.
